# The Empire of Affectivity: Qualitative Evidence of the Subjective Orgasm Experience

**DOI:** 10.3390/bs14030171

**Published:** 2024-02-23

**Authors:** Pablo Mangas, Mateus Egilson da Silva Alves, Ludgleydson Fernandes de Araújo, Juan Carlos Sierra

**Affiliations:** 1Mind, Brain and Behavior Research Center (CIMCYC), University of Granada, 18011 Granada, Spain; pablomangas@ugr.es; 2Department of Pós-Graduação em Psicologia, Universidade Federal do Delta do Parnaíba, Parnaíba 64202-020, Brazil; mateusegalves@gmail.com (M.E.d.S.A.); ludgleydson@gmail.com (L.F.d.A.)

**Keywords:** subjective orgasm experience, multidimensional model of the subjective orgasmic experience, Orgasm Rating Scale, sexual relationships, masturbation, qualitative evidence

## Abstract

The subjective orgasm experience (SOE) refers to its perception and/or assessment from a psychological viewpoint. Few works have approached this construct from a qualitative perspective and have never taken a consolidated theoretical model as a reference. This study aims to provide qualitative validity evidence to the Multidimensional Model of Subjective Orgasmic Experience, derived from the Orgasm Rating Scale (ORS), to qualitatively address SOE in the contexts of sexual relationships and solitary masturbation, analyzing the terms self-generated by individuals and examining the coincidence with the semantic descriptions of orgasm proposed by the ORS. Four hundred Spanish adults aged 18 to 64 years participated. The Technique of Free Association of Words was applied, and prototypical, frequency, and similitude analyses were performed. A similar description was observed concerning the terms generated in both contexts, with a higher frequency and intensity in the context of sexual relationships. In the context of solitary masturbation, negative orgasmic descriptions were evoked. Participants were able to elicit the vast majority of ORS adjectives, with Affective being the most notable dimension, followed closely by Rewards, especially in masturbation. Most of the adjectives were evoked simultaneously with those of the Affective, with “pleasurable” standing out as the most predominant one. This work provides qualitative evidence to the SOE study, ratifying the semantic composition of the ORS and thus endorsing the Multidimensional Model of Subjective Orgasmic Experience as a good theoretical model from which to continue studying the subjective orgasmic experience.

## 1. Introduction

Orgasm can be defined as the moment of maximum sexual pleasure, characterized by rhythmic contractions of the perineal organs and accompanied by changes in the cardiovascular and respiratory systems, as well as the release of sexual tension [1]. This pleasurable sensation generates an alteration of consciousness that produces well-being and satisfaction [2]. Traditionally, the study of orgasm has focused on the evaluation of its presence/absence and/or difficulty in obtaining it [3,4,5]. In recent years, there has been an interest in approaching orgasm from the assessment of its subjective experience [6,7,8,9,10,11,12], referring to its perception and evaluation at a psychological level.

The study of the subjective orgasm experience (SOE) has its origins in the notion that orgasm is not a single-dimensional phenomenon but rather that, in addition to its sensory dimension, there is a cognitive-affective component [13,14]. Subsequently, Mah and Binik [8] extended this multidimensional approach of SOE to different contexts, populations, and evolutionary stages.

Taking as a reference the instruments developed to assess SOE, this construct has been operationalized in several ways. First, we find the approach proposed by the Orgasm Rating Scale (ORS) [15,16]. The ORS aims to examine SOE based on adjectives that rate orgasm, attending to individual subjectivity, both in the context of sexual relationships and solitary masturbation [16]. The creation of the ORS required a review of the literature that compiled descriptions of orgasms to generate a set of 141 adjectives [17], a number that was later progressively reduced. The Spanish adaptation has 25 adjectives organized into four dimensions: Affective, Sensory, Intimacy, and Rewards [15]. Another approach to SOE is the one proposed by Dubray et al. [18], who developed the Bodily Sensations of Orgasm Questionnaire, which characterizes the orgasmic experience by attending only to specific genital and extragenital (bodily and physiological) sensations. Moreover, the Orgasmometer [19] also measures subjective orgasmic intensity using a graded visual color scale (from white to red) [19,20]. Finally, the most recent approach to SOE is proposed by the Orgasmic Perception Questionnaire (OPQ) [11], proposing five dimensions: ecstasy, contractions, well-being, power, and sensations. Unlike the ORS, this instrument is made up of complex definitions and not mere adjectives, as is the case with the ORS.

The relevance of SOE lies in its association with variables related to sexual health, such as sexual satisfaction [6,9], relationship satisfaction [9], or erotophilia, sexual desire, arousal, and sexual functioning [19,20,21,22,23,24,25]. Mah and Binik [8] studied SOE in the contexts of sexual relationships and solitary masturbation and found differential manifestations. It has been evidenced that SOE depends on the context in which orgasm occurs, being more intense in the context of sexual relationships than in the context of solitary masturbation [10,26].

The Multidimensional Model of Subjective Orgasmic Experience (MMSOE), proposed by Mah and Binik [8] and based on the ORS, is presented as a consolidated theoretical model that conceptualizes SOE based exclusively on the psychological appraisal of orgasm. Arcos-Romero et al. [27] proposed a tetra-dimensional structure comprising (1) the Affective dimension, which encompasses the emotions experienced during orgasm; (2) the Sensory dimension, relating to the perception of physiological changes; (3) the Intimacy dimension, reflecting the intimate aspect of the orgasmic experience; and (4) the Rewards dimension, denoting the consequences derived from orgasm. The MMSOE constitutes a theoretical reference in the study of orgasm, allowing its dimensions to account for the individuality of the orgasmic experience. This model has clinical applicability in the promotion of sexual health, given the associations between its dimensions and sexual satisfaction [6,9].

The MMSOE has evidence of validity since its dimensions have been associated with different measures of sexual excitation, both in the context of sexual relationships [27] and in the context of solitary masturbation [28]. The fact that the ORS—the scale on which the MMSOE is based—has been validated in the Spanish population, both in the context of heterosexual [15] and homosexual [22] relationships and in the context of solitary masturbation in heterosexual [29] and LGB [10] individuals, justifies its use as a theoretical background for the present work.

Few studies have qualitatively addressed the subjective orgasm experience. As an exception, Opperman et al. [30] explored the meanings young people gave to orgasm and pleasure during sexual encounters with a partner, generating themes that showed the great complexity and contradictory meanings associated with the orgasmic experience. To date, there are no qualitative studies that address SOE under the MMSOE framework. Therefore, the present study aims to approach SOE from a qualitative perspective by attending to the contexts in which orgasm occurs (i.e., sexual relationships and solitary masturbation). For this purpose, (a) we will examine the adjectives self-generated by individuals to describe their orgasmic experiences in both contexts, (b) we will analyze the frequency with which these self-generated adjectives coincide with the semantic description of orgasm proposed by the ORS, and (c) we will explore the co-occurrence between these self-generated adjectives that describe the orgasmic experience.

## 2. Materials and Methods

### 2.1. Type of Study

This is an exploratory, descriptive, qualitative study based on cross-sectional data and non-probabilistic sampling.

### 2.2. Participants

The participants were 400 Spanish adults aged 18 to 64 years (29.30 ± 7.60), considering the legal age of majority in Spain (18 years). Among them, there was the presence of heterosexual men (*n* = 81), non-heterosexual men (*n* = 82), heterosexual women (*n* = 112), and non-heterosexual women (*n* = 125). We established the criteria for inclusion as having had orgasmic experiences in the last three months, both in the context of sexual relationships and in the context of solitary masturbation. A total of 87.2% had a university education (*n* = 349), 12.3% had secondary education (*n* = 49), and 0.5% had primary education (*n* = 2). Overall, 68% were in a couple relationship (*n* = 272), and 32% were single (*n* = 128). Regarding first sexual experiences, participants reported a mean age of first sexual relationship (oral, vaginal, and/or anal) of 17.07 ± 2.86 years and first masturbation of 12.8 ± 3.61 years.

### 2.3. Instruments

A Sociodemographic and Sexual History Questionnaire was employed, which collected information on sex, age, educational level, nationality, sexual orientation, relationship status, age of first sexual relationship (oral, vaginal, and/or anal), and first masturbation. The Technique of Free Association of Words proposed by Vergès [31] was used as a word abduction instrument [32], in hierarchical evocation and sensitive to the investigated content from an inducing stimulus [33,34,35]. In the present study, two inducing stimuli were applied in relation to SOE: “orgasm in sexual relationships” and “orgasm in solitary masturbation”, asking participants to indicate the first five words that originated in their thoughts in relation to them. Specifically, both questions were formulated as follows: “Now try to recall as best you can the most recent orgasm you experienced during sexual relationship with another person. You may include any sexual activity in which you had an orgasm with a sexual partner present [...]. What words would you use to describe that orgasm? Please write down the first five words that come to your mind. You should place them in order of importance and separated by commas”. The wording regarding the second inducing stimulus was analogous but referred to the orgasm experienced during solitary masturbation.

### 2.4. Procedure

The evaluation of participants was conducted online, a common procedure for assessing sexual behaviors [36], via social media, distribution channels, and email lists, using the free software LimeSurvey^®^ (version 5.6.31), located on the servers of the University of Granada, among Spanish adults. The dissemination was carried out between February and May 2023. All participants were informed of the purpose and voluntary nature of the study, as well as the implications of their participation. The anonymity and confidentiality of the responses were guaranteed, and their use was limited to purposes related to scientific research and dissemination. No personal data were required. All participants accepted an informed consent form. The approximate time required to complete the questionnaire was 15 min. To avoid fraudulent responses, an alphanumeric CAPTCHA based on a random arithmetic operation was used at the beginning of the survey. The data were thoroughly reviewed to eliminate cases with inconclusive responses or anomalous patterns. The present study was previously approved by the Human Research Ethics Committee of the University of Granada (Ref. 2308/CEIH/2021).

### 2.5. Analyses

The data obtained via the inducing stimuli were tabulated in an Open Office spreadsheet, and their content was hierarchized following the same order as the participants’ evocations. Next, we proceeded to lemmatization, a procedure for grouping lexical terms with the same semantic structures, adopting the most frequent semantic class as the criterion of prevalence [37].

The data were analyzed using the IRaMuTeQ software (Interface de R pour les Analyses Multidimensionnelles de Textes et de Questionnaires) version 0.7 alpha 2, used by R Studio version 3.6.2, which allows obtaining statistical data from qualitative material from texts, questionnaires, and transcripts, among others [34,38]. This free software is capable of integrating qualitative analyses into quantitative matrices [39,40].

Initially, a prototypical analysis was performed for each of the stimuli under study (orgasm in sexual relationships and orgasm in masturbation). This strategy uses a spreadsheet of hierarchical data to categorize them according to their significant elements, allowing to obtain as a resulting product four quadrants (upper quadrants, called central core and first periphery, and lower quadrants, called zone of contrast and second periphery), considering their frequencies and mean evocations [35,37,41]. The central core represents high-frequency words with a high level of importance; the first periphery offers high-frequency words but with lower importance; the zone of contrast includes lower-frequency words but with a high level of importance; and the second periphery represents low-frequency words with a low level of importance. This was followed by a frequency analysis for the evoked adjectives that coincided with those proposed by the ORS. Finally, by means of similitude analysis, these adjectives coinciding with those of the ORS were analyzed. Via this procedure, the proximity between the elements of the analyzed content is revealed via co-occurrence indices, making it easier to see how attractive the adjectives are to appear close to each other in the same environment or how much they repel each other [42,43].

## 3. Results

### 3.1. Prototypical Analysis

Prototypical analysis categorizes elements based on their frequency, determined by significantly increased usage of specific words or expressions, alongside their perceived importance, indicated by the initial evocation they elicit [31]. In the first place, the Technique of Free Association of Words was used for each of the two stimuli: “orgasm in sexual relationships” and “orgasm in masturbation”, adopting the minimum frequency criterion [37] as the retention criterion for their inclusion in the quadrants (in the case of this study, from two evocations). The mean recall order indicator “Range” considered words less than or equal to 2.88 in the case of the first stimulus (Table 1) and 2.87 in the case of the second stimulus (Table 2) to have a low recall order, indicating that they do not appear in the first evocation positions. The first quadrant (central core) shows the words that were most frequently evoked (pleasurable, intense, satisfying, etc., in the case of orgasm in sexual relationships; and pleasurable, intense, quick, etc., in masturbation), which may indicate greater group cohesion and stability to the particular stimulus. We can infer that the adjectives associated with this zone are very meaningful to the group under the induction of each of the stimuli. The second quadrant, in contrast, contains words from the first periphery (relaxing, pleasant, fun, etc., in the case of orgasm in sexual relationships; and satisfying, short, liberating, etc., in the case of orgasm in masturbation) which, although evoked less frequently, do so more easily, fundamentally in the first positions (Range > 2.88/2.87). These are complementary elements to the central core, which could have a transitional nature towards it in the long term. The third and fourth quadrants contain the elements of the zone of contrast and of the second periphery. The frequency of evocation is low compared to the other quadrants (less frequent elements), and they provide fewer representative descriptions if the whole group of participants is taken into account. However, the manifestation of these elements may also indicate transitions to the previous zones. This information is also relevant since it denotes the presence of more individual elements that, on a larger scale, could be representative of the group, highlighting the abundant presence of clearly negative elements associated with the orgasmic experience in the context of masturbation (Table 2), nonexistent in the context of sexual relationships (incomplete, anxious, unpleasurable, dirty, mechanic, cold, sad, tired, repetitive, and unnecessary); see Table 1 and Table 2.

### 3.2. Frequency Analysis

This strategy was implemented with the aim of verifying the evocations directly associated with the descriptions proposed by the ORS, considering the whole set of evoked terms and the number of participants mentioning them. This analysis allows the expanded verification of the frequencies (multiple frequencies) obtained from the planning (matrix) of the data, independently of the Range (evocation order) [44]. See Table 3. Specifically, for the sexual relationships context, participants evoked all twenty-five adjectives of the ORS, with the exception of two (shooting and spreading). For the solitary masturbation context, participants were able to evoke all but four of the ORS adjectives (tender, flooding, rising, and spreading). The adjective “spreading” was the only one that was not evoked on any occasion by any participant to describe orgasm in either context.

The multiple frequencies showed that the terms “pleasurable”, “relaxing”, and “satisfying” coincide in being the most frequently cited by the participants (belonging to the Affective, Rewards, and Affective ORS dimensions, respectively). Moreover, when the set of items of each dimension is analyzed, a similar fact is observed: the Affective dimension is the most notorious in both contexts, especially in the sexual relationships context, with its adjectives being cited by more than 90% of the participants. This was followed by the Rewards, Intimacy, and Sensory dimensions, in that order, as the most expressed in both contexts. However, in the context of masturbation, the Rewards factor has a higher prominence, achieving almost the same representation in terms of the Sum of the dimensional load as the Affective dimension, something that does not occur in the context of sexual relationships, where although Affective and Rewards occupy the first and second positions, there is a great difference in evocation between them. The Intimacy and Sensory dimensions, although less evoked, were both more expressed in the context of sexual relationships.

### 3.3. Similitude Analysis

This procedure is useful to identify connections between intertextual linguistic forms by visualizing how their content is structured [45]. This technique is applied to textual structures by checking how they approach or distance themselves from a central axis from which the links originate [42,46]. In this study, we decided to examine term co-occurrence, looking at how evocations are connected to each other from ORS-related inducing terms. In reference to evocations of orgasm in sexual relationships, the central axis with the highest expressivity was “pleasurable”, being associated with great intensity to “satisfying”, “exciting”, “exploding”, and “relaxing”, and from where the other connections originate. Three of these five adjectives belong to the Affective dimension, including the central axis, which corroborates the data from the prototypical and multiple frequencies analysis, showing that this factor seems to be very significant for the group. In addition, the more distant terms show some connection with the central axes (there are no isolated items); Figure 1.

When analyzing the evocations based on the induction of the term orgasm in masturbation, similar associations were observed in the context of sexual relationships, so “pleasurable” was the central and most expressive axis, and “satisfying”, “exciting”, and “relaxing” were also closely linked to it. This shows a trend quite similar to that found in the context of sexual relationships, where the dimension that gained the most prominence was Affective. On this occasion, occurrences that are not linked to any axis also stand out, such as “loving” appearing in isolation and “euphoric” and “wild” showing co-occurrence only between them, although both appear isolated in relation to the central axis. This dispersion may indicate that the group is able to express greater dynamism or idiosyncrasy in this scenario compared to that of sexual relationships; Figure 2.

## 4. Discussion

The aim of this study was to provide qualitative evidence for the study of the subjective orgasm experience (SOE). To date, very few studies have qualitatively addressed aspects related to the orgasmic experience [30], and we have no precedents from the perspective of a consolidated theoretical model such as the one proposed based on the ORS, specifically its validation in the Spanish population [15,29]. The MMSOE is a theoretical reference in the study of orgasm, having evidence of validity due to the correlation of its dimensions with different measures of sexual excitation [27,28], also showing clinical applicability given the associations between SOE and sexual satisfaction [6,9]. For this study, the tetra-dimensional orgasm conceptualization proposed by the MMSOE based on the ORS was used as a theoretical framework.

Due to the contextual nature of SOE (sexual relationships vs. solitary masturbation), the first of our objectives was to examine the adjectives self-generated by individuals to describe their orgasmic experiences in both contexts. In the prototypical analysis, first of all, we observed a similar descriptive tendency in both contexts, especially in the first adjectives corresponding to the central core (the most frequently mentioned in both contexts were “pleasurable” and “intense”). Although the general tendency is for frequencies to be higher in the context of sexual relationships, this fact denotes a sort of horizontality between both scenarios, something already reported in previous evidence [10,47], which pointed out that, while SOE is generally intense in both contexts, it is more so in the context of sexual relationships. This is corroborated precisely by the second of the adjectives with the highest frequency in the two contexts (intense), which is more evoked in the sexual relationships’ context than in the masturbation one.

Considering other topographic parameters, we observed ambiguous manifestations, especially in the context of sexual relationships. For example, with regard to duration, the terms “short” (the only one with these characteristics in the central core), “quick”, and “brief” appear in this context; however, the one most frequently evoked is “long” (belonging to the first periphery). These four adjectives also appeared in the context of solitary masturbation, being, in this case, the most evoked “quick”, to which “lasting” and “slow” are added in the zone of contrast, solidly representing only a very small group of participants. This leads us to think that in the context of sexual relationships, there is a certain heterogeneity among the participants in terms of self-perceived time invested in reaching orgasm, while in the context of masturbation, there is more uniformity in stating that orgasm is generally obtained more rapidly. This is consistent with studies that have examined orgasmic latency, in which women take nearly twice as long to reach orgasm with a partner as they do with solitary masturbation [48], a context in which orgasm is typically reached in 7–8 min. This may be explained by the fact that clitoral stimulation is more direct in masturbation versus shared sexual encounters, resulting in faster orgasm achievement [49,50], or in their ability to experience multiple orgasms, a fact that, although it also exists in men [51], is much more documented in women [52,53]. In the case of men, this could be explained by the tendency to instrumentalize orgasm to obtain something derived from it [10,22], such as relaxing or falling asleep. Future studies should qualitatively examine SOE in relation to factors such as gender to shed more light on this issue. On the other hand, in relation to intensity—beyond the adjective “intense” discussed above—we find many more clues in the context of sexual relationships that invite us to think that orgasms in this context are more intense: “heavy”, “brutal”, “animal”, and “wild”. Despite this, there is still a certain ambiguity since the term “soft” also appears in the central core, although it is evoked almost half as often as “heavy”. In the context of masturbation, we hardly found references to this parameter, except for “powerful” with few evocations, which again translates into a more intense experience of orgasm in the context of sexual relationships than in masturbation [10,47].

As we expected, taking into account the characteristics of each of the contexts, in the context of sexual relationships, we observed some terms that denote the dyadic nature of the orgasmic experience (e.g., complicity, communication), while in the context of masturbation, this type of evocation was not found, but rather the individual nature of the behavior (i.e., solitary). Observing some adjectives in the context of solitary masturbation, especially “necessary”, “desired”, and “wanted”, we can interpret that the group gave prominence to orgasms obtained via this behavior (there seems to be an apparent balance between both contexts), without the presence of adjectives that denote compensatory characteristics of orgasm in sexual relationships. While we are unaware of the frequency with which each person engages in both scenarios and, especially, the satisfaction derived from the orgasms experienced in each of them, taking as a reference the complementary vs. compensatory hypothesis of masturbation with respect to sexual relationships [54,55], we are inclined to think that for the sample of our study, both behaviors are complementary. Future studies should take into account additional variables to clarify this question.

Additionally, while in both contexts it is possible to observe elements representative of a finalistic erotica (focused more on the results than on the development or course of the sexual behavior itself), this occurred to a greater extent in the context of masturbation (i.e., relaxing, peaceful, de-stressing, liberating, and relief), which is similar to the findings of Muñoz-García et al. [10]. In this context, moreover, we observed the presence of elements that suggest a certain mechanism, habituation, and ordinariness of this behavior (i.e., routine, handy, daily, mechanic, known, and repetitive), something that is consistent with previous literature that has associated masturbation behavior with a propensity to boredom [56,57] and fatigue [58]. The descriptions “routine” and “repetitive” should be taken into account, as both could produce a detriment in sexuality [59]. Another notable difference between both contexts is the presence only in the masturbation context of cognitive elements, referring to imagination or fantasies (i.e., fanciful and imagination), being this consistent with the works that point out that masturbation behavior is associated with the ability to fantasize sexually, both in men [60] and in women [61]. Furthermore, a relationship has been found between a positive attitude towards sexual fantasies with a positive attitude towards masturbation behavior [60].

From this analysis, we also offer something that is not contemplated by the different models that have operationalized the orgasmic experience: negative elements associated with orgasm. It is noteworthy that only in the context of solitary masturbation -especially in the third and fourth quadrants—there is a presence of clearly negative descriptors: “boring”, “incomplete”, “anxious”, “unpleasurable”, “dirty”, and “unnecessary”. This fact has also been recently identified by Panzeri et al. [11], who suggested the lack of negative items associated with orgasm during the elaboration phase of the Orgasmic Perception Questionnaire (OPQ). That these terms appear only when describing orgasms obtained via masturbation could imply that a percentage of individuals experience orgasms under that condition in an aversive or unpleasant way or that they directly manifest reticence towards the behavior itself, something that has been found to be associated with lower orgasm experiences [60] and even as an element that hinders orgasmic capacity per se [62]. Future focus should be placed especially on the analysis of negative emotions associated with orgasmic experiences in this solitary context.

Furthermore, similarities were observed between many of the adjectives evoked by the participants and those included in the ORS. We found that all the adjectives in the Spanish version of the ORS [15] were generated spontaneously by the respondents, with the exception of two in the context of sexual relationships (shooting and spreading) and four in the context of masturbation (tender, flooding, rising, and spreading). The only adjective that was not evoked on any occasion was “spreading”. This is contrary to previous evidence, as existing studies that ranked adjectives based on the ORS indicate that “spreading” is reported by 68% of heterosexual men and 78.6% of heterosexual women [7] and by 76 and 73.3% of bisexual men and women, respectively, as well as by 63.8% and 76% of gays and lesbians, respectively [12]. However, according to the study by Cervilla et al. [29] in the context of masturbation, this adjective was one of those that showed the greatest variability in terms of intensity of the ORS in this context, with 58% reporting it as very representative and 42% of their sample reporting it as very unrepresentative. Since our study did not provide clues and favored free association, it is likely that this adjective, although well known, is little used in Spanish. Nevertheless, we found that other synonymous terms were evoked [63] that have not been part of this analysis because they are not adjectives derived directly from the ORS (e.g., intense or passionate).

The frequency analysis reflected a parallelism between both contexts (i.e., sexual relationships and solitary masturbation): the three most frequent adjectives were “pleasurable”, “relaxing”, and “satisfying”. In the analysis by dimensions, we observed another identical feature in both scenarios: the most evoked dimensions were Affective, followed by Rewards, Intimacy, and Sensory. The fact that the Affective dimension was the most evoked by participants, especially in the context of sexual relationships (where more than 90% of participants evoked some adjective corresponding to it) coincides with previous studies [17,47] in which SOE was more intense in the context of relationships. Muñoz-García et al. [10], comparing both contexts, pointed out differences in the Affective dimension, reporting greater intensity in the context of sexual relationships. Additionally, this result also ratifies the findings of the few dyadic studies that have addressed SOE [9,64], in which the Affective dimension is so strongly loaded that it seems to mask the others. Moreover, in these studies, the Affective dimension is the only one associated, both in its actor and partner effect, with sexual satisfaction in male couples [9], as well as the only one to show a negative association between both contexts [64], meaning that, in same-sex male and female couples, a high intensity of SOE in its Affective dimension in the context of masturbation is associated with lower levels of global SOE in sexual relationships.

These results are also identical to those derived from the ORS-based adjective rankings, where in the context of sexual relationships, the first five adjectives that top the list, and therefore best describe orgasm, are all from the Affective dimension, both in heterosexual [7] and bisexual, gay, and lesbian individuals [12]. Another noteworthy aspect is the recompensing nature of orgasm in the context of masturbation (the dimensional load of the Rewards dimension is practically identical to Affective in this context). This, in addition to being in line with our previous results, which already showed a tendency to describe orgasms obtained in this context using terms associated with the consequences (e.g., relaxing, de-stressing, or relief), again demonstrates the rewarding nature of orgasm in masturbation [10], which highlights a further tendency to instrumentalize orgasms in this context, focusing on the consequences derived from it.

Finally, we observed how the evoked terms are connected to the inducing terms related to the ORS. In line with the previous results, there is evidence that the dimension that occupies the central core is Affective. Specifically, it is appreciated that the adjectives derived from the ORS are linked in some way with “pleasurable” (central term) and with “satisfying” and “exciting”. This translates into participants describing their orgasms in both contexts with terms mostly derived from the Affective dimension. This fact replicates what was found in quantitative studies in the context of sexual relationships, in which “pleasurable” was precisely the term that best described people’s orgasmic experiences, occurring both in heterosexual men and women [7] and in bisexual women, gays and lesbians (in bisexual men it is the second most intense, after “satisfying”) [12]. When analyzing the adjectives individually, in general, there is a strong parallelism between the results obtained in the present study and previous quantitative evidence (see Arcos-Romero and Sierra [7] and Sierra et al. [12] to discover the complete rankings). In the context of masturbation, something similar occurred: Cervilla et al. [29] reported “pleasurable” as the third ORS item describing orgasm most intensely, only surpassed by “elated” and “satisfying”, the latter also being one of the most representative in the present study.

We observed two notable differences between the contexts of sexual relationships and solitary masturbation. First, although in both scenarios, the presence of the term “relaxing” (Rewards dimension) is powerful, it is more prominent in the solitary masturbation scenario, as the number of terms linked to it is higher when evoking descriptions of orgasm in solitary masturbation. This is also in line with the rest of our analyses, where, although the Affective dimension is the most salient, in the case of masturbation, it is also closely followed by the Rewards dimension, the only one in which quantitative evidence has shown that SOE is more intense in the context of masturbation [10]. On the other hand, in the context of sexual relationships, all the evoked terms are linked to the central axis, which implies a certain homogeneity and consensus among the descriptions. In contrast, in the context of masturbation, there are some terms not linked to the central axis, appearing dispersed (“loving” from the Intimacy dimension) or dispersed, but with co-occurrence between them (“euphoric” and “wild”, both from the Sensory dimension). This result is also consistent with that reported in the study by Cervilla et al. [29], in which these items showed much ambiguity, being scored very intensely by some individuals and not very intensely by others. This means that the participants in the present study showed more heterogeneity when describing their orgasms in the context of solitary masturbation, which could be due to the fact that masturbation, being a private and idiosyncratic sexual behavior, does not require as much consensus, coordination and adaptation to social norms as a sexual encounter in which another person is present, so that perhaps in these solitary experiences people could be displaying more creativity and fewer restrictions. This particularity of masturbation may be explained by the fact that, via its practice, it favors learning and body self-knowledge [65] and can be used for various purposes, such as obtaining pleasure and relaxation, among others [61,66].

The aim of this study was to provide qualitative evidence for the study of SOE, something that had not been carried out from the perspective of a consolidated theoretical model, such as the MMSOE. The above findings show great similarity with all previous quantitative evidence analyzing SOE. However, this study is not without limitations that affect the generalizability of the results. Participants were selected by incidental sampling and were mostly young and highly educated. In addition, the instruments were disseminated via social media (which makes it difficult for people without access to them to participate). Although this work was intended to provide descriptive and exploratory evidence on the differences between contexts, future studies should analyze SOE, also under the contextual focus (sexual relationships vs. masturbation), but analyzing more in-depth sociodemographic factors such as gender, sexual orientation, or age, to see if previous quantitative evidence continues to be replicated, and also collect data on drug or medication consumption, factors that may affect the orgasmic response. Another interesting approach would be to incorporate mixed methodologies in the study of the subjective orgasmic experience. Future work should also study SOE from other theoretical paradigms of operationalization of this construct, as well as explore it in people with orgasmic difficulties. Additionally, it would be essential to pay attention in the future to the negative descriptions of the orgasmic experience found in this study, something that, to date, has not been captured by the different approaches to conceptualizing SOE. Finally, we would like to emphasize that orgasm should not undeniably be considered the best indicator of sexual or relationship satisfaction. As pointed out by authors such as Fahs [67] and Thorpe et al. [68], the absence of orgasm does not necessarily translate into people not having pleasurable relationships, just as the presence of orgasm does not make an encounter unquestionably ideal [9].

## 5. Conclusions

The present study provides qualitative evidence to the study of SOE, ratifying the semantic structure of the ORS and the MMSOE derived from it, consolidating it as an adequate model from which to continue studying subjective orgasmic experience, both in the context of sexual relationships and masturbation. Our results highlight the prominence of adjectives belonging to the Affective dimension when describing SOE in both contexts, in addition to the protagonism of the Rewards dimension, especially in the context of masturbation. Negative descriptions associated with orgasm exclusively in masturbation also emerged, as well as more heterogeneity of responses in this context.

## Figures and Tables

**Figure 1 behavsci-14-00171-f001:**
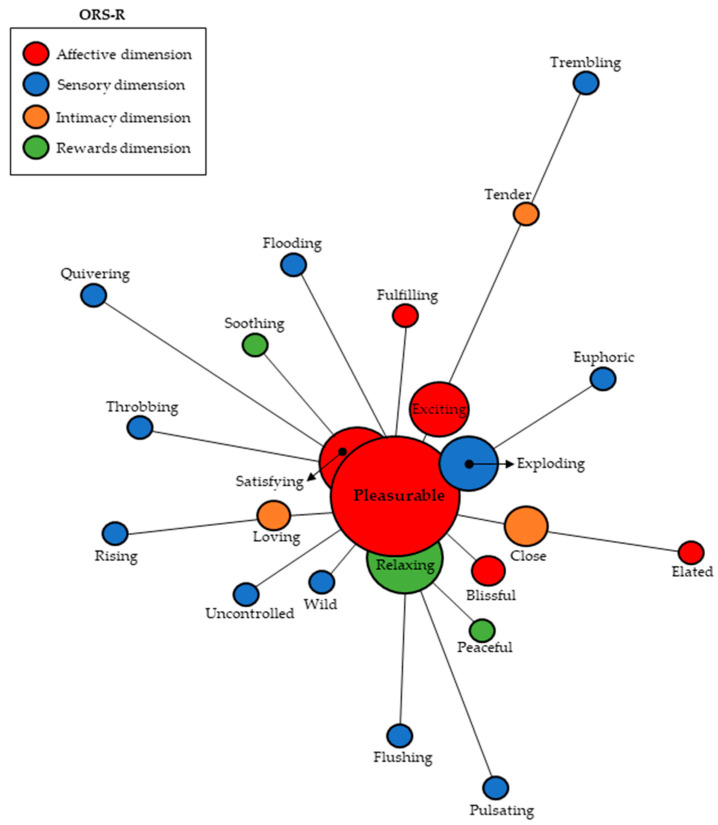
Analysis of similitude based on co-occurrence for adjectives coinciding with the Orgasm Rating Scale in the context of sexual relationships.

**Figure 2 behavsci-14-00171-f002:**
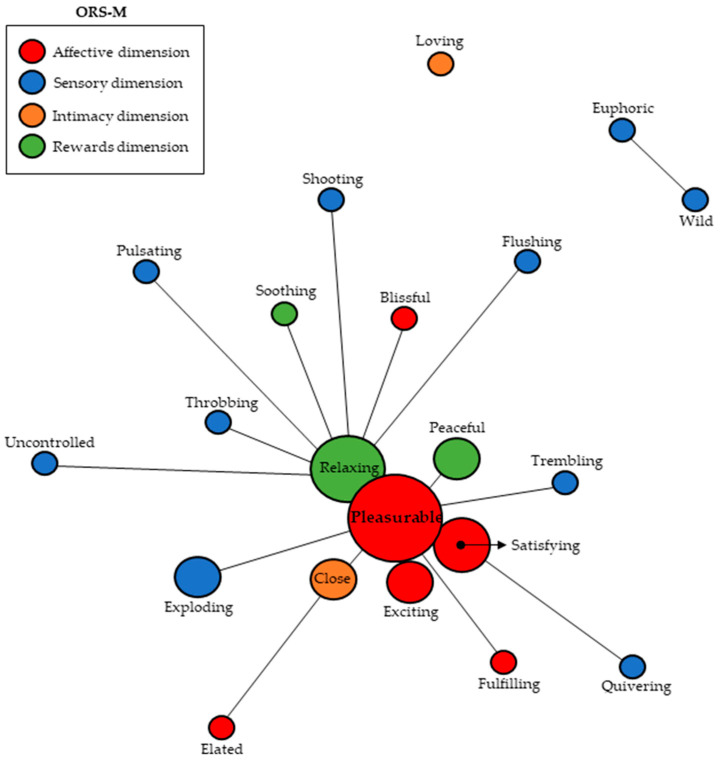
Analysis of similitude based on co-occurrence for adjectives coinciding with the Orgasm Rating Scale in the context of solitary masturbation.

**Table 1 behavsci-14-00171-t001:** Prototypical analysis with the stimulus “Orgasm in sexual relationships”.

	Range ≤ 2.88			Range > 2.88		
Average Frequency	Central Core			First Periphery		
	Evocations	*f*	Range	Evocations	*f*	Range
	Placentero/Pleasurable	228	2.1	Relajante/Relaxing	72	3.4
	Intenso/Intense	195	1.9	Agradable/Pleasant	47	3.3
	Satisfactorio/Satisfying	61	2.8	Divertido/Fun	35	3.3
	Excitante/Exciting	49	2.8	Íntimo/Close	32	3.3
	Liberador/Liberating	42	2.8	Deseado/Desired	30	3.4
	Explosivo/Exploding	36	2.8	Largo/Long	30	3
≥9.66	Corto/Short	26	2.3	Bonito/Beautiful	24	3.4
	Fuerte/Heavy	20	2.5	Bueno/Good	24	3
	Maravilloso/Blissful	15	2.6	Amoroso/Loving	22	4.1
	Pasional/Passionate	15	2.7	Rápido/Quick	20	3.3
	Suave/Soft	11	2.3	Feliz/Happy	18	3.8
	Éxtasis/Ecstasy	11	1.8	Gustoso/Tasty	17	3.6
	Increíble/Incredible	10	2.7	Húmedo/Wet	15	3.3
	Eléctrico/Electric	10	2	Caliente/Hot	15	3.6
Average Frequency	Zone of Contrast			Second Periphery		
	Evocations	*f*	Range	Evocations	*f*	Range
	Breve/Brief	6	2.5	Apasionado/Passionate	9	3.8
	Brutal/Brutal	6	2.5	Reconfortante/Soothing	9	3.4
	Rico/Delicious	5	2	Romántico/Romantic	9	4.1
	Complicidad/Complicity	5	2	Salvaje/Wild	8	3
	Completo/Complete	5	2.6	Tranquilizante/Peaceful	8	3.6
	Nuevo/New	4	2.5	Gratificante/Fulfilling	8	3.4
<9.66	Progresivo/Progressive	4	2.5	Morboso/Lustful	8	3.6
	Pasión/Passion	4	2	Emocionante/Exciting	7	3.4
	Eufórico/Euphoric	4	2.8	Conexión/Connection	7	3.4
	Extático/Ecstatic	3	2.3	Profundo/Deep	7	3.6
	Caluroso/Warm	3	2.7	Cómplice/Complicit	7	3.4
	Animal/Animal	2	2.5	Seguro/Safe	6	3
	Saciante/Satiating	2	2.5	Estimulante/Stimulating	6	4.3
	Comunicación/Communication	2	2.5	Sensual/Sensual	5	3.2

**Table 2 behavsci-14-00171-t002:** Prototypical analysis with the stimulus “Orgasm in masturbation”.

	Range ≤ 2.87			Range > 2.87		
Average Frequency	Central Core			First Periphery		
	Evocations	*f*	Range	Evocations	*f*	Range
	Placentero/Pleasurable	163	2.4	Satisfactorio/Satisfying	60	3.2
	Intenso/Intense	125	2	Corto/Short	41	3
	Rápido/Quick	114	2.3	Liberador/Liberating	39	2.9
	Relajante/Relaxing	103	2.8	Agradable/Pleasant	37	3
	Íntimo/Close	34	2.6	Solitario/Solitary	31	3.6
	Tranquilizante/Peaceful	25	2.8	Bueno/Good	21	3.1
≥8.96	Excitante/Exciting	25	2.7	Divertido/Fun	19	3.3
	Rutinario/Routine	18	2.6	Fácil/Easy	18	3.3
	Desestresante/De-stressing	18	2.6	Caliente/Hot	16	3.2
	Necesario/Necessary	16	2.4	Aburrido/Boring	15	3.1
	Explosivo/Exploding	16	1.9	Gustoso/Tasty	14	3.1
	Largo/Long	16	2.7	Cómodo/Comfortable	13	3.8
	Breve/Brief	15	2.4	Fantasioso/Fanciful	10	2.9
	Deseado/Desired	11	2.3	Normal/Normal	10	3.5
Average Frequency	Zone of Contrast			Second Periphery		
	Evocations	*f*	Range	Evocations	*f*	Range
	Buscado/Wanted	8	2.1	Sucio/Dirty	8	3.8
	Alivio/Relief	7	2.3	Gratificante/Fulfilling	8	3.9
	Simple/Simple	6	2.8	Maravilloso/Blissful	8	3
	Práctico/Handy	6	2.8	Increíble/Incredible	8	3.4
	Duradero/Lasting	5	2.5	Mecánico/Mechanic	8	3.2
	Imaginación/Imagination	5	2.4	Húmedo/Wet	7	3.3
<8.96	Lento/Slow	5	2	Potente/Powerful	7	3.7
	Incompleto/Incomplete	4	2.8	Frío/Cold	7	3.3
	Controlado/Controlled	4	2.2	Triste/Sad	6	4.5
	Cotidiano/Daily	4	2.8	Conocido/Known	6	3.3
	Espontáneo/Spontaneous	4	2	Cansado/Tired	5	4.6
	Palpitante/Pulsating	3	2.7	Repetitivo/Repetitive	4	3.8
	Ansioso/Anxious	3	2.3	Funcional/Functional	4	3.5
	Displacentero/Unpleasurable	2	1.5	Innecesario/Unnecessary	4	4.8

**Table 3 behavsci-14-00171-t003:** Frequencies and percentages of occurrence of adjectives coinciding with the Orgasm Rating Scale.

		ORS-R					ORS-M		
	Adjectives	Frequency	% of Total	% of Rows	Rank	Rank	Frequency	% of Total	% of Rows
A	Pleasurable (*placentero*)	228	5.83	57.0	1	1	163	4.24	40.25
R	Relaxing (*relajante*)	72	1.84	18.0	2	2	103	2.68	25.75
A	Satisfying (*satisfactorio*)	61	1.56	15.25	3	3	60	1.56	15.0
A	Exciting (*excitante*)	49	1.25	12.25	4	6	25	0.65	6.25
S	Exploding (*explosivo*)	36	0.92	9.0	5	7	16	0.42	4.0
I	Close (*íntimo*)	32	0.82	8.0	6	4	34	0.88	8.5
I	Loving (*amoroso*)	22	0.56	5.5	7	16	1	0.03	0.25
A	Blissful (*maravilloso*)	15	0.38	3.75	8	10	8	0.21	2.0
R	Soothing (*reconfortante*)	9	0.23	2.25	9	11	5	0.13	1.25
S	Wild (*salvaje*)	8	0.2	2.0	10	17	1	0.03	0.25
R	Peaceful (*tranquilizante*)	8	0.2	2.0	11	5	25	0.65	6.25
A	Fulfilling (*gratificante*)	8	0.2	2.0	12	9	8	0.21	2.0
S	Throbbing (*vibrante*)	5	0.13	1.25	13	8	9	0.23	2.25
S	Uncontrolled (*incontrolable*)	4	0.1	1.0	14	19	1	0.03	0.25
I	Tender (*tierno*)	4	0.1	1.0	15	-	-	-	-
S	Euphoric (*eufórico*)	4	0.1	1.0	16	15	2	0.05	0.5
S	Trembling (*tembloroso*)	3	0.08	0.75	17	12	3	0.08	0.75
S	Flooding (*desbordante*)	2	0.05	0.5	18	-	-	-	-
S	Flushing (*sofocante*)	2	0.05	0.5	19	18	1	0.03	0.25
A	Elated (*gozoso*)	2	0.05	0.5	20	14	3	0.08	0.75
S	Quivering (*estremecedor*)	2	0.05	0.5	21	21	1	0.03	0.25
S	Rising (*creciente*)	1	0.03	0.25	22	-	-	-	-
S	Pulsating (*palpitante*)	1	0.03	0.25	23	13	3	0.08	0.75
S	Shooting (*desbocado*)	-	-	-	-	20	1	0.03	0.25
S	Spreading (*efusivo*)	-	-	-	-	-	-	-	-
	Dimensions	Frequency	% of Total	% of Rows	Σ	Σ	Frequency	% of Total	% of Rows
	Affective	363	9.27	90.75	60.5	44.5	267	6.95	66.25
	Sensory	68	1.74	17	6.18	3.8	38	1.01	9.5
	Intimacy	58	1.48	14.5	19.33	17.5	35	0.91	8.75
	Rewards	89	2.27	22.25	29.67	44.33	133	3.46	33.25

Note. A = Affective; S = Sensory; I = Intimacy; R = Rewards; Σ = sum of the dimensional load (frequency divided by the number of items of each dimension).

## Data Availability

The data presented in this study are available upon request from the corresponding author. The data are not publicly available due to privacy.

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
