# Peer review of "The Empire of Affectivity: Qualitative Evidence of the Subjective Orgasm Experience"

_behavsci, 2024, doi:10.3390/bs14030171_

Round 1

Reviewer 1 Report

Comments and Suggestions for Authors

The manuscript “The Empire of Affectivity: Qualitative Evidence of the Subjective Orgasm Experience” focuses on an interesting topic. The manuscript is well written and clear, so I think it should be accepted for publication. However, I'll leave a comment even if only to make a provocation to the authors.

1. In fact, the introduction is clear, highlighting, as required, the relevance of the study to science and closing the section with the objectives of the study.

2. The Materials and Methods section has five subsections (Type of study, Participants, Instruments, Procedure, and Analyses), all very complete and clear.

3. The results section, which contains several tables and figures, is also very clear.

4. The discussion section is also quite good, highlighting some limitations and presenting suggestions for future studies, as is required.

However, I'd like to point out that, given that you have enough socio-demographic information to carry out an analysis from a gender perspective, I was surprised that you didn't do it, merely presenting it as a suggestion for future studies in the discussion. If you have all the information on the sex or gender of the participants, why did you choose not to do this analysis?

In fact, although I'm not an expert on orgasm experience, from the knowledge I have, it seems fundamental to explore the differences and similarities of sex and gender in relation to this topic. Nevertheless, I leave it up to you to decide whether, or not, to include this analysis.

Author Response

The authors thank the reviewers for their efforts in reviewing the manuscript. Their comments and suggestions allow us to significantly improve the work.

The following is a response to each of the reviewers' comments and suggestions.

The manuscript “The Empire of Affectivity: Qualitative Evidence of the Subjective Orgasm Experience” focuses on an interesting topic. The manuscript is well written and clear, so I think it should be accepted for publication. However, I'll leave a comment even if only to make a provocation to the authors.

>>General response: First of all, we would like to thank you, on behalf of our research team, for taking the time to review our manuscript. We are very glad to read that it meets your expectations, that you find it interesting and the potential you see in it for publication in this journal. We will be glad to respond to your comments in the next sections.

  1. In fact, the introduction is clear, highlighting, as required, the relevance of the study to science and closing the section with the objectives of the study.
  2. The Materials and Methods section has five subsections (Type of study, Participants, Instruments, Procedure, and Analyses), all very complete and clear.
  3. The results section, which contains several tables and figures, is also very clear.
  4. The discussion section is also quite good, highlighting some limitations and presenting suggestions for future studies, as is required.

>>Response: We thank you very much again for finding these sections of our manuscript appropriate. We are pleased to have clearly transmitted all the logic and organization of the different sections of the research.

However, I'd like to point out that, given that you have enough socio-demographic information to carry out an analysis from a gender perspective, I was surprised that you didn't do it, merely presenting it as a suggestion for future studies in the discussion. If you have all the information on the sex or gender of the participants, why did you choose not to do this analysis?

In fact, although I'm not an expert on orgasm experience, from the knowledge I have, it seems fundamental to explore the differences and similarities of sex and gender in relation to this topic. Nevertheless, I leave it up to you to decide whether, or not, to include this analysis.

>>Response: Thank you very much for bringing this issue to light. On this occasion, as this is the first qualitative approach to orgasmic experience from the consolidated theoretical paradigm proposed by the Multidimensional Model of Subjective Orgasmic Experience (MMSOE), derived from the Orgasm Rating Scale (ORS), we merely wanted to observe whether its dimensionality and semantic richness of its composition were qualitatively justified. To this end, as a first approximation, we only decided to analyze contextually (sexual relationships vs. masturbation).

Taking into account that there are definitely differences by gender and sexual orientation in this psychosexual dimension, it would have been very interesting to present the results attending to different sociodemographic variables such as those you propose, but we believe that, in the case of this manuscript, this would have drastically increased the length of the work. To give an example, presenting the results by gender would have implied doubling the number of tables and figures in the manuscript, with the consequent extension of the Discussion. For this reason, our perspective focuses on developing this line of research in a more staggered manner.

Since we are starting this line of qualitative investigation from our research team, precisely the next steps will be centered on the direction you have suggested. Given that with this study we have confirmed the dimensionality of the theoretical model, and we have effectively verified that it replicates with certain accuracy the quantitative evidence that we already knew, after this exploratory and descriptive work we will begin another phase of study in which we will attend to sociodemographic aspects, as follows:

  1. To explore qualitatively, in the context of sexual relationships uniquely, the role of participants' gender, to observe whether this evidence replicates the quantitative results of previous studies.
  2. To examine the same, but in the context of solitary masturbation.
  3. To investigate how the sexual orientation factor affects the results obtained, first in the context of sexual relationships.
  4. Then in the context of masturbation.

The analytical strategy and the results of this type of study are so complex in terms of interpretation and discussion that the next steps will be, once the previous evidence is known (in this study we have provided qualitative evidence of validity to the model), to analyze some sociodemographic variables such as those mentioned above, separately in each of the two contexts.

We will be delighted that, in one way or another, the next manuscripts in this line of research will reach your hands in the future so that you can continue to appreciate the advances in this field of study.

Thank you again for your time, the interesting discussion generated and for the value you have been able to appreciate in this facet of human sexuality.

Reviewer 2 Report

Comments and Suggestions for Authors

The current manuscript sought to verify the MMSOE utilizing a qualitative approach. The Technique of Free Association of Words was used to explore the theoretical framework and support the semantic words used in the ORS. This was an interesting article to read and it adds to the literature on orgasmic experiences, further supporting the notion that orgasms are a multifaceted experience. I have a broad question about the analytic method used.

This comment is in the context of my unfamiliarity of the particular analysis technique...I don't understand how the table headings (i.e., 'central core' and 'first periphery') are connected to the results/type of analysis. More explanation would be helpful. 

Broadly, I think the jargon associated with The Technique of Free Association of Words might be overly used to describe the results, which ultimately gets in the way of understanding the findings and its broad implications. 

Author Response

The authors thank the reviewers for their efforts in reviewing the manuscript. Their comments and suggestions allow us to significantly improve the work. We also thank the Editor for the opportunity to resubmit a corrected version of the article.

The following is a response to each of the reviewers' comments and suggestions.

The current manuscript sought to verify the MMSOE utilizing a qualitative approach. The Technique of Free Association of Words was used to explore the theoretical framework and support the semantic words used in the ORS. This was an interesting article to read and it adds to the literature on orgasmic experiences, further supporting the notion that orgasms are a multifaceted experience. I have a broad question about the analytic method used.

>>General response: First of all, we would like to thank you, on behalf of our research team, for taking the time to review this manuscript. We are very glad to read that you found it interesting and that you consider that it could be a valuable addition to the previous literature, confirming that orgasmic experience should be considered from a multifaceted perspective. We will then be happy to discuss with you any suggestions or questions that may have arisen.

This comment is in the context of my unfamiliarity of the particular analysis technique... I don't understand how the table headings (i.e., 'central core' and 'first periphery') are connected to the results/type of analysis. More explanation would be helpful.

Broadly, I think the jargon associated with The Technique of Free Association of Words might be overly used to describe the results, which ultimately gets in the way of understanding the findings and its broad implications.

>>Response: Thank you very much for your comment. On this occasion, we prefer to answer these two questions in a combined and global manner. We recognize that this type of analysis technique can be unfamiliar and even somewhat complex, but we would be delighted to hear that you are having a friendly introduction to it. A description of each of these four quadrants (within the first analysis, the prototypical one) appears in the results section of the manuscript. We have included an explicit reference to the 'central core' and we have also marked the rest in red, to make it more understandable. We have also included a definition of this analysis at the beginning of the section.

Generally considered, this is the usual table format offered by IRAMUTEQ when performing prototypical analysis. The writing format is very similar to other manuscripts where this technique is used. We will be happy to give you a summary of the meaning of each of the quadrants below:

  • Central core: high frequency elements and high levels of importance.
  • First Periphery: high frequency, but lower importance.
  • Zone of contrast: lower frequency, but high importance level.
  • Second Periphery: low frequency and low importance level.

We confirm that in the new version of the manuscript this information has also been added. As indicated in the text, the quadrants are not immutable, which means that perhaps the inclusion of a word in one of them may have a transitional nature in the future, and become part of another quadrant, since the representations that we make fluctuate during our lifetime.

For further familiarization, we strongly recommend reading Mengzhen et al. (2023) and Giacomozzi et al. (2023), two recent, really interesting papers, in which this particular analysis is also applied.

Giacomozzi, A. I., Rozendo, A., da Silva Bousfield, A. B., Leandro, M., Fiorott, J. G., & da Silveira, A. (2023). COVID-19 and elderly females—A study of social representations in Brazil. Trends in Psychology, 31(2), 429-445. https://doi.org/10.1007/s43076-021-00089-9

Mengzhen, L., Lim, D. H. J., Berezina, E., & Benjamin, J. (2024). Navigating love in a post-pandemic world: Understanding young adults’ views on short-and long-term romantic relationships. Archives of Sexual Behavior, 53(2), 497-510. https://doi.org/10.1007/s10508-023-02738-9

If it is still not clear enough or if you consider that some variable or table heading should be explained in more detail in the manuscript, do not hesitate to ask us and we will be happy to improve it.

Thank you again for your review, your valuable time and for considering our work as an interesting contribution to this field of study.

Reviewer 3 Report

Comments and Suggestions for Authors

Dear Authors.

Thank you for inviting me to review this manuscript. In the attached document you have the proposed suggestions for improvement. 

Best regards.

Author Response

The authors thank the reviewers for their efforts in reviewing the manuscript. Their comments and suggestions allow us to significantly improve the work. We also thank the Editor for the opportunity to resubmit a corrected version of the article.

The following is a response to each of the reviewers' comments and suggestions.

I appreciate the invitation to review this study, which aimed to provide qualitative evidence for the examination of subjective orgasmic experience. The comprehension of sexual health is significantly enhanced by adopting a qualitative approach to the study of subjective orgasmic experiences. The incorporation of this aspect not only enhances the approach to sexual health but also contributes to a broader understanding of health in general. Additionally, it validates the Multidimensional Model of Subjective Orgasmic Experience as a robust theoretical framework for future research, offering a holistic perspective to enhance the quality of both sexual and emotional life. Congratulations on your choice of subject matter. After reviewing the article, I will proceed to make a series of considerations for each section of the manuscript, always with a constructive view and potential for improvement.

>>General response: First of all, we thank you, on behalf of our research team, for the thorough review of our manuscript and the valuable time invested in our work. We are very pleased to read that you consider it important to provide this area of research with qualitative evidence, as well as for the positive and constructive feedback received. We would be honored to discuss with you in the following sections any questions or concerns that may have arisen during the reading of our work.

Title and keywords. Nothing to add, the title is appropriate for the research being undertaken. The keywords are appropriate.

>>Response: Thank you again for your time.

Abstract. The summary is correctly focused. However, I would like to make two points.

  1. No acronyms or abbreviations may be included in the abstract. Please remove them.
  2. Please state the objective of this research more clearly.

>>Response: Thank you again for bringing these issues to our attention. In the case of our abstract, we must recognize that we have only eliminated the acronym referring to the MMSOE. We understand your concern, but on this occasion, we think that the most logical thing to do is to keep them, since the length limit of the abstract was 200 words (according to the journal guidelines) and we were already allowed to submit it slightly exceeding this limit. Not using acronyms would increase the length even more. If, however, you consider it more appropriate not to use them, bearing this information in mind, we would be happy to do so. Regarding the objective, and again for reasons of length, we have introduced the primary objective of the study briefly. The incorporated information can be reviewed in the new version of the manuscript.

Introduction. I congratulate you on a clear and well-written introduction that introduces the topic of study in an appropriate manner. I have no comments on this.

>>Response: Thank you again. We are glad that you find it an appropriate Introduction.

Materials and methods. The methodology allows the study to be adequately replicated. However, I propose a number of suggestions for improvement.

  1. Avoid providing results of the socio-demographic variables (age, sex...) of the sample in the methodology. If possible, include this information in the results. When referring to the mean±standard deviation, indicate it in the following way mean±standard deviation to make it clearer and easier to read and understand. Example. The mean age of the population was 54.2±7.1years.

>>Response: Thank you very much for the comments on these points. In the case of the present manuscript, we decided to incorporate the sociodemographic information in the Materials and methods section, and not in the Results, because sociodemographic data were not used for the analyses. These data were not the objective of the present study. Since this information is relatively simple and brief, and was not used, we opted for this option. Knowing this logic, if you still think it is more appropriate to report this information in the Results section, please let us know and we will be happy to do so.

In reference to the format in which you have recommended us to report "mean±standard", we inform you that this logic has been incorporated in the new version of the manuscript. Thank you again.

  1. Did you collect data on sexual orientation under "non-heterosexual"?

>>Response: Thank you very much for bringing this issue to light. In the case of the present study, a question based on the well-known Kinsey scale was used to find out the sexual orientation of the participants. Although this variable was not used in this study for the analyses, we wanted to have as balanced a sample as possible, so we included people with different sexual orientations. Within the "non-heterosexual" group there are lesbians, gays and bisexuals.

  1. Did they consider the age of majority in Spain (+18 years) as an inclusion criterion? If yes, please indicate. If not, were they able to obtain samples from minors (with the authorization of their parents or legal guardians)?

>>Response: Thank you very much for asking this question, and we regret not having made it explicit in the previous version of the manuscript. Effectively, we only consider the participation of people with legal age of majority in Spain (18 years of age or older). We have modified this aspect in the new version of the manuscript.

  1. The instruments selected for data collection are in line with the objective of the study. The procedure is well described and can be reproduced without complications. I congratulate you.

>>Response: Thank you very much for considering appropriate the selection of instruments that we made, as well as the description of the procedure of our study.

 Results. Overall, the presentation of the results is clear and structured. There is a detailed description of the quadrants and their interpretation in the analysis, as well as the use of tables and figures to visualize the data. I have nothing to add.

>>Response: Thank you very much for your comment. We are very grateful that you found the clarity, structure and details of this section, as well as the tables and figures provided, to be optimal and understandable.

Discussion. The discussion is well structured and clearly written. There is a thorough discussion, which situates the findings of this study with previous evidence. I would like to ask you a question, as a future line of research, do you think that a mixed methodology (in the same research) could bring more conclusive results to the topic of study? I congratulate you on the discussion.

>>Response: We thank you very much for finding the Discussion of our manuscript appropriate. The question you pose is very interesting, and indeed it is something we plan to do in the future in this line of research. We believe that adopting a mixed approach may offer more innovative results. One possible way to continue with this type of study that we have been thinking about for some time is to use the Spanish versions of the Orgasm Rating Scale in the context of sexual relationships and solitary masturbation, in conjunction with some open-ended response or word evocation question. It would be very interesting, for example, to establish groups based on the orgasmic intensity of the participants, to account for the representations of a sample of the Spanish population with high orgasmic intensity vs. low orgasmic intensity. We believe that a study of these characteristics, although complex, could offer promising results. We have included in the section on future lines the possibility of carrying out mixed research. Thank you again.

Conclusions. I have no comments; this section is appropriate for study.

>>Response: Thank you again. We are glad that you found this section appropriate.

References. I would like to congratulate you on the appropriate ageing of references. Please revise your formatting to conform to journal guidelines. Specifically, I am referring to the year of publication of references 64 and 65.

>>Response: Thank you very much for your comment about the references used in our manuscript. We carefully reviewed the format of the references and we believe that these two were found to be correct. Number 64 is a manuscript submitted for publication, and number 65 is not an article, but a book. In both cases the year of publication is not in bold, if that is what you are referring to.

We direct you to the 'Instructions for Authors' of the journal, where you can check the concreteness of these references, which would correspond to: 'Unpublished materials intended for publication' (number 64) and 'Books and Book Chapters' (number 65).

https://www.mdpi.com/journal/behavsci/instructions

We would like to take this opportunity to reiterate our gratitude for the debate generated, for the meticulous review of our manuscript, as well as for the suggestions for improvement. We believe that thanks to your collaboration, our work now has a higher level of quality.

Reviewer 4 Report

Comments and Suggestions for Authors

Thanks to the editors for inviting me to review the manuscript. Congratulations to the authors for their efforts. 

The manuscript is very well done. I would like to make only a few suggestions for improvement. 

The difference between male and female participation is important, therefore, I recommend either to include this aspect in the limitations or to present the results by men and women. Also, it would have been interesting to know if the participants consumed any type of drug or medication that could affect orgasm. 

Author Response

The authors thank the reviewers for their efforts in reviewing the manuscript. Their comments and suggestions allow us to significantly improve the work. We also thank the Editor for the opportunity to resubmit a corrected version of the article.

The following is a response to each of the reviewers' comments and suggestions.

Thanks to the editors for inviting me to review the manuscript. Congratulations to the authors for their efforts. The manuscript is very well done. I would like to make only a few suggestions for improvement.

>>General response: Thank you very much for your time in reviewing this manuscript. We are very glad to read your compliments and that you found our research appropriate. We will be happy to read and discuss your suggestions.

The difference between male and female participation is important, therefore, I recommend either to include this aspect in the limitations or to present the results by men and women. Also, it would have been interesting to know if the participants consumed any type of drug or medication that could affect orgasm.

>>Response: Thank you very much for bringing this topic to light. We agree with you that men and women differ in many facets of sexuality, including the subjective orgasmic experience (SOE).

As for gender, in the version of the manuscript that you already reviewed, this same question appears in future lines of study. Since the aim of the present study was to provide only descriptive and exploratory evidence of SOE, we decided to run the analyses only in the contexts in which orgasms occurs (sexual relationships vs. masturbation). We have to inform you that this is the first work in a new line of research in which we will continue to analyze SOE qualitatively.

 The following steps will be aimed at analyzing, in each of the two contexts separately, the role played by some sociodemographic variables (especially gender and sexual orientation). In a manuscript of this type, analyzing separately by gender, for example, would considerably increase the length (the tables, figures and, consequently, the Discussion would be duplicated). For this reason, in future research we will focus on only one of the two contexts, taking into account these sociodemographic factors. Therefore, this reasoning made us make explicit what you comment in the future lines section, rather than in limitations.

On the other hand, what we find really interesting is the second of your proposals. We strongly believe that collecting information about drug or medication consumption could provide valuable information, as these are certainly factors that affect the orgasmic response. We confirm that we have incorporated this question in the future lines of research of the new version of the manuscript.

Thank you very much again for your words and help us to improve our work.